# Maternal Cardiovascular Risk Assessment 3-to-11 Years Postpartum in Relation to Previous Occurrence of Pregnancy-Related Complications

**DOI:** 10.3390/jcm8040544

**Published:** 2019-04-20

**Authors:** Ilona Hromadnikova, Katerina Kotlabova, Lenka Dvorakova, Ladislav Krofta

**Affiliations:** 1Department of Molecular Biology and Cell Pathology, Third Faculty of Medicine, Charles University, 10000 Prague, Czech Republic; katerina.kotlabova@lf3.cuni.cz (K.K.); lenka.dvorakova@lf3.cuni.cz (L.D.); 2Institute for the Care of the Mother and Child, Third Faculty of Medicine, Charles University, 14700 Prague, Czech Republic; ladislav.krofta@upmd.eu

**Keywords:** BMI (body mass index), blood pressure, cardiovascular diseases, gestational hypertension, fetal growth restriction, heart rate, preeclampsia, risk, serum markers, QRISK^®^2 risk score, waist circumference

## Abstract

The aim of the present study was to assess the long-term outcomes of women 3-to-11 years postpartum in relation to the previous occurrence of pregnancy-related complications such as gestational hypertension (GH), preeclampsia (PE) and fetal growth restriction (FGR). Body mass index (BMI), waist circumference values, the average values of systolic (SBP) and diastolic (DBP) blood pressures and heart rate, total serum cholesterol levels, serum HDL (high-density lipoprotein) cholesterol levels, serum LDL (low-density lipoprotein) cholesterol levels, serum triglycerides levels, serum lipoprotein A levels, serum CRP (C-reactive protein) levels, plasma homocysteine levels, serum uric acid levels, individual and relative risks of having a heart attack or stroke over the next ten years were compared between groups (50 GH, 102 PE, 34 FGR and 90 normal pregnancies) and correlated with the severity of the disease with regard to clinical signs (25 PE without severe features, 77 PE with severe features), and delivery date (36 early PE, 66 late PE). The adjustment for potential covariates was made, where appropriate. At 3–11 years follow-up women with a history of GH, PE regardless of the severity of the disease and the delivery date, PE without severe features, PE with severe features, early PE, and late PE had higher BMI, waist circumferences, SBP, DBP, and predicted 10-year cardiovascular event risk when compared with women with a history of normotensive term pregnancy. In addition, increased serum levels of uric acid were found in patients previously affected with GH, PE regardless of the severity of the disease and the delivery date, PE with severe features, early PE, and late PE. Higher serum levels of lipoprotein A were found in patients previously affected with early PE. The receiver operating characteristic (ROC) curve analyses were able to identify a substantial proportion of women previously affected with GH or PE with a predisposition to later onset of cardiovascular diseases. Women with a history of GH and PE represent a risky group of patients that may benefit from implementation of early primary prevention strategies.

## 1. Introduction

Pregnancy-associated hypertension provokes long-standing metabolic and vascular changes that might augment the global risk of diabetes mellitus, cardiovascular diseases, cerebrovascular diseases, as well as kidney diseases, later in life [1,2,3]. Recent studies gave clear evidence that preeclampsia (PE) or eclampsia had been associated with the risk for latter onset of metabolic syndrome, hypertension, atherosclerosis, ischemic heart disease, congestive heart failure, stroke, and deep venous thrombosis [4,5,6,7,8,9]. This increase in risk ranges from a doubling of risk in all cases to an eight-to-ninefold increase in women with early PE requiring the delivery before 34 weeks of gestation [10,11,12,13,14]. 

In addition, women with gestational hypertension (GH) have also been observed to have an increased risk for ischemic heart disease, myocardial infarcts, heart failure, and ischemic stroke [3]. Parallel, either women bearing growth retarded fetuses or women delivering infants with low birth weight were identified to be at a significant risk for subsequent development of ischemic heart disease as well [15]. 

Recent findings revealed that inadequate adaptation of the maternal cardiovascular system on the physiologically increasing requirements during pregnancy had been associated with the onset of late PE and late fetal growth restriction (FGR). Late PE or late FGR may occur as a consequence of secondary placental dysfunction caused by the inability of the maternal cardiovascular system to cope with the significant volume load and metabolic demands of an advanced or overgrown pregnancy [16,17]. Nevertheless, early PE and early FGR are usually caused by primary placental insufficiency, which is present from the beginning of gestation [16,18].

The aim of the current prospective study was to evaluate associations between pregnancy-related complications such as gestational hypertension, preeclampsia and/or fetal growth restriction and maternal cardiovascular risk.

We compared BMI, waist circumference values, the average values of systolic (SBP) and diastolic (DBP) blood pressures and heart rate, total serum cholesterol levels, serum HDL (high-density lipoprotein) cholesterol levels, serum LDL (low-density lipoprotein) cholesterol levels, serum triglycerides levels, serum lipoprotein A Lp(a) levels, serum CRP (C-reactive protein) levels, plasma homocysteine levels, serum uric acid levels, individual and relative risks of having a heart attack or stroke over the next ten years postpartum (3 to 11 years after the delivery) among women with normal and complicated course of gestation. 

A risk of having a heart attack or stroke over the next ten years was estimated in relation to pregnancy outcome using the QRISK^®^2-2016 Web Calculator (https://qrisk.org/2016/index.php) (QRISK^®^—a registered trademark of the University of Nottingham and EMIS, Nottingham, NG7 2RD, United Kingdom; website and risk engine built by ClinRisk Ltd., Leeds, West Yorkshire, LS1 2TW, United Kingdom). The QRISK^®^2 score contains many of the traditional risk factors included in Framingham score (such as age, cholesterol/HDL ratio, systolic blood pressure, diabetes and smoking status, blood pressure treatment), but also contains important additional risk factors (ethnicity, family history of premature coronary heart disease in a first degree relative under the age of 60, body mass index, occurrence of rheumatoid arthritis, chronic kidney disease, and atrial fibrillation). 

Classic cardiovascular disease risk factors and QRISK^®^2 CVD score were analysed in relation to the subtypes and the severity of previous pregnancy-related complications with respect to the occurrence of PE without severe features (PE w/o SF), PE with severe features (PE w/SF), early PE (requiring the delivery before 34 weeks of gestation), and late PE (the delivery after 34 weeks of gestation). The influence of additional factors was taken under consideration during the interpretation of the data. We made adjustments for potential covariates, where appropriate, including current maternal age, BMI and parity, the time after the birth (in months), the oral contraceptive use status (ex-user, current user, non-user), the smoking status (ex-smoker, smoker, non-smoker), the average values of systolic and diastolic blood pressures, and hypertension on the treatment (yes, no). 

The identification of young women at high risk of cardiovascular disease may provide an opportunity for early personalized follow-up and prevention [10]. The severity of the increased cardiovascular disease risk has been recognized by the American Heart Association, which now recommends that a pregnancy history be part of the evaluation of cardiovascular risk in women [19,20].

The originality of this study lies in the complexity of the chosen approach. A whole range of clinical parameters and serum biomarkers associated with an increased cardiovascular risk was analysed and compared between the groups of normal and complicated pregnancies with emphasis on subgroups of pregnancy-related complications (GH, PE w/o SF, PE w/SF, early PE, late PE, and FGR). We focused on the examination of BMI, waist circumference values, the average values of systolic and diastolic blood pressures and heart rate, total serum cholesterol levels, serum HDL cholesterol levels, serum LDL cholesterol levels, serum triglycerides levels, serum Lp(a) levels, serum CRP levels, plasma homocysteine levels, serum uric acid levels, individual and relative risks of having a heart attack or stroke over the next ten years. 

## 2. Results

### 2.1. The Clinical Characteristics of Normal and Complicated Pregnancies

The clinical characteristics of normal and complicated pregnancies are presented in Table 1.

### 2.2. Impact of A History of Gestational Hypertension and Preeclampsia Irrespective of the Severity of the Disease on Maternal Cardiovascular Risk

At three–11 years follow-up, women with a history of GH and PE, regardless of the severity of the disease and the delivery date, had higher BMI both before and after the adjustment of the data for appropriate covariates (GH: *p* < 0.001, *p* = 0.002^A^, PE: *p* < 0.001, *p* = 0.001^A^), waist circumferences (GH: *p* < 0.001, *p* = 0.001^A^, PE: *p* < 0.001, *p* = 0.001^A^), SBP (GH: *p* < 0.001, *p* < 0.001^A^, PE: *p* < 0.001, *p* = 0.029^A^), DBP (GH: *p* < 0.001, *p* < 0.001^A^, PE: *p* < 0.001, *p* = 0.002^A^), and predicted 10-year cardiovascular event risk (GH: *p* < 0.001, *p* < 0.001^A^, PE: *p* < 0.001, *p* < 0.001^A^) when compared with women with a history of normotensive term pregnancy (Table 2, Table S1).

The ROC curve analyses revealed significantly higher BMI (GH: 42.0%, 74.0% ^A^; PE: 27.45%, 53.06% ^A^), waist circumferences (GH: 42.0%, 70.0% ^A^; PE: 32.35%, 54.08% ^A^), SBP (GH: 54.0%, 62.0% ^A^; PE: 46.86%, 55.1% ^A^), DBP (GH: 51.0%, 60.0% ^A^; PE: 39.71%, 47.96% ^A^), and predicted 10-year risk of having a heart attack or stroke (GH: 30.0%, 74.0% ^A^; PE: 26.37%, 55.1% ^A^) in a substantial proportion of mothers previously affected with GH or PE at 10.0% FPR both before and after the adjustment of the data for appropriate covariates (Table 3, Table S2).

Furthermore, women with a prior exposure to GH (*p* = 0.004) and PE regardless of the severity of the disease and the delivery date (*p* = 0.005) had higher levels of serum uric acid, but only if the data were not adjusted for appropriate covariates. The ROC curve analysis showed increased levels of serum uric acid in 36.53% patients with a history of GH and 28.51% patients with a history of PE at 10.0% FPR (Table 2 and Table 3, Appendix A).

### 2.3. Impact of a History of Preeclampsia without and with Severe Features on Maternal Cardiovascular Risk

Both groups of patients affected with either PE w/o SF or PE w/SF showed at 3–11 year follow-up as before so after the adjustment of the data for appropriate covariates significantly BMI (PE w/o SF: *p* = 0.021, *p* = 0.021^A^, PE w/SF: *p* < 0.001, *p* = 0.015^A^), waist circumferences (PE w/o SF: *p* = 0.010, *p* = 0.029^A^, PE w/SF: *p* < 0.001, *p* = 0.013^A^), DBP (PE w/o SF: *p* = 0.014, *p* = 0.036^A^, PE w/SF: *p* < 0.001, *p* < 0.001^A^), and predicted 10-year cardiovascular event risk (PE w/o SF: *p* = 0.008, *p* = 0.003^A^, PE w/SF: *p* < 0.001, *p* < 0.001^A^), or displayed a trend toward higher SBP after the adjustment of the data for appropriate covariates (PE w/o SF: *p* = 0.001, *p* = 0.055^A^, PE w/SF: *p* < 0.001, *p* = 0.121^A^) when compared with the control group of women with a history of normal pregnancy (Table 4, Table S3).

The sensitivity at 10.0% FPR before the adjustment of the data for appropriate covariates for BMI (PE w/o SF: 20.0%, PE w/SF: 29.87%), waist circumferences (PE w/o SF: 28.0%, PE w/SF: 33.77%), SBP (PE w/o SF: 40.0%, PE w/SF: 49.09%), DBP (PE w/o SF: 24.0%, PE w/SF: 44.81%), and predicted 10-year risk of having a heart attack or stroke (PE w/o SF: 19.87%, PE w/SF: 28.48%) was slightly higher for mothers after pregnancy affected with PE w/SF when compared to those ones affected with PE w/o SF (Table 5, Table S4).

On the other hand, the sensitivity at 10.0% FPR after the adjustment of the data for appropriate covariates for BMI (PE w/o SF: 62.5%; PE w/SF: 56.76%), waist circumferences (PE w/o SF: 58.33%; PE w/SF: 55.41%), SBP (PE w/o SF: 62.5%; PE w/SF: 56.76%), DBP (PE w/o SF: 45.83%; PE w/SF: 50.0%), and predicted 10-year risk of having a heart attack or stroke (PE w/o SF: 58.33%; PE w/SF: 54.05%) was approximately equal between mothers after PE w/o SF and PE w/SF pregnancy.

Moreover, women previously exposed to PE w/SF (*p* = 0.001) had higher levels of serum uric acid, but only if the data were not adjusted for appropriate covariates. The ROC curve analysis showed increased levels of serum uric acid in 28.82% patients with a history of PE w/SF at 10.0% FPR (Table 4 and Table 5, Appendix A).

### 2.4. Impact of A History of Early and Late Preeclampsia on Maternal Cardiovascular Risk

Women with a history of early PE and late PE had, at longitudinal follow-up study both before and after the adjustment of the data for appropriate covariates, significantly higher BMI (early PE: *p* < 0.001, *p* = 0.003^A^, late PE: *p* = 0.034, *p* = 0.031^A^), waist circumferences (early PE: *p* < 0.001, *p* = 0.011^A^, late PE: *p* = 0.006, *p* = 0.018^A^), DBP (early PE: *p* < 0.001, *p* < 0.001^A^, late PE: *p* < 0.001, *p* = 0.004^A^), and predicted 10-year cardiovascular event risk (early PE: *p* < 0.001, *p* < 0.001^A^, late PE: *p* = 0.009, *p* = 0.005^A^), or displayed a trend toward higher SBP after the adjustment of the data for appropriate covariates (early PE: *p* < 0.001, *p* = 0.099^A^, late PE: *p* < 0.001, *p* = 0.021^A^) when compared with women with a history of uncomplicated pregnancies (Table 6, Table S5).

The sensitivity at 10.0% FPR before the adjustment of the data for appropriate covariates for BMI (early PE: 38.89%, late PE: 21.21%), waist circumferences (early PE: 50.0%, late PE: 22.73%), SBP (early PE: 62.78%, late PE: 38.18%), DBP (early PE: 54.17%, late PE: 31.82%), and predicted 10-year risk of having a heart attack or stroke (early PE: 41.67%, late PE: 18.03%) was substantially higher for mothers previously exposed to early PE than in mothers with a prior exposure to late PE (Table 7, Table S6).

In parallel, after the adjustment of the data for appropriate covariates, it was found that at 10.0% FPR a markedly higher proportion of mothers exposed to early PE had higher BMI (early PE: 67.65%; late PE: 48.44%), waist circumferences (early PE: 64.71%; late PE: 50.0%), SBP (early PE: 70.59%; late PE: 46.88%), DBP (early PE: 64.71%; late PE: 34.38%), and predicted 10-year risk of having a heart attack or stroke (early PE: 73.53%; late PE: 51.56%) than the mothers with a prior exposure to late PE (Table 7, Table S6). 

In addition, ANOVA analysis indicated higher levels of serum uric acid in women previously exposed to early PE (*p* = 0.001) and late PE (*p* = 0.018), but only for the unadjusted data. The ROC curve analysis showed increased serum levels of uric acid in 42.86% patients with a history of early PE and 20.91% patients with a history of late PE at 10.0% FPR (Table 6 and Table 7, Appendix A).

Besides, increased serum levels or a trend toward increased serum levels of lipoprotein A were found in patients previously affected with early PE before and after the adjustment of the data for appropriate covariates (*p* = 0.037, *p* = 0.192^A^). The ROC curve analysis was able to identify higher levels of Lp(a) in 31.43% (unadjusted data) and 81.82% (adjusted data) women affected with early PE (Table 6 and Table 7, Appendix A).

## 3. Discussion

This longitudinal follow-up study revealed that women with a prior exposure to GH and PE had higher BMI, waist circumferences, systolic blood pressure, diastolic blood pressure, serum levels of uric acid, and increased relative risks of having a heart attack or a stroke over the next ten years when compared with women with a history of normal course of gestation. 

With regard to the severity of PE with respect to the degree of clinical signs and delivery date, higher BMI, waist circumferences, systolic blood pressure, diastolic blood pressure, and an increased predicted 10-year risk of having a heart attack or stroke were identified in all subgroups of patients with a history of PE (PE w/o SF, PE w/SF, early PE, and late PE). Nevertheless, increased postpartum serum levels of uric acid were found only in patients with a history of PE w/SF regardless of the fact if it occurred before or after 34 weeks of gestation. In addition, increased postpartum serum levels of lipoprotein A were observed entirely in patients previously exposed to early PE. 

Women with GH (unadjusted data 1.966, adjusted data 1.984) and/or PE (unadjusted data 1.617, adjusted data 1.578) were identified to be at an increased risk of a heart attack or a stroke. Women previously affected with PE w/SF seemed to have even more unfavourable cardiovascular risk profile when compared with women with a history of PE w/o SF (unadjusted data 1.744 vs. 1.228, adjusted data 1.553 vs. 1.201). Similarly, a history of early PE was associated with a higher QRISK^®^2 score than a history of late PE (unadjusted data 2.436 vs. 1.171, adjusted data 2.099 vs. 1.162).

This finding may support previous findings of other groups that GH and/or PE (merged together as gestational hypertensive disorders or studied separately) are more likely to be associated with postpartum higher BMI [21,22,23,24,25,26,27,28,29], waist circumferences [21,26,30], systolic blood pressure [21,22,26,27,28,29,31,32], diastolic blood pressure [21,22,26,27,28,29,31,32,33], an increased cardiovascular risk over the next ten years [22,29,34,35,36,37], or an increased cardiovascular risk over the next 30 years [22,38] assessed by using a modified Framingham risk score or NORRISK 2 (a Norwegian risk model for acute cerebral stroke and myocardial infarction.

The current study produced similar findings to previous follow-up studies, in which the association between mild PE (PE w/o SF) [21,39] or severe PE (PE w/SF) [21,39] and higher BMI [21,39], waist circumferences [21], systolic blood pressure [21,39], and diastolic blood pressure [21,39] was reported.

Our data are also consistent to a certain extent with the data resulting from several previous studies [23,31,40], where significantly higher serum levels of uric acid were found in patients with a prior exposure to hypertensive pregnancy disorders or PE only. Hyperuricemia is associated with cardiovascular disease risk factors and cardiovascular diseases, often being a predictor of incident events [41,42,43]. Assessment of renal function is important in the management of patients with kidney diseases or pathologies affecting renal function. One woman involved in the follow-up study indicated to suffer from chronic kidney disease (nephrotic syndrome). We identified some women with presence of risk factors predisposing for chronic kidney disease (1 PE with haematuria, 2 GH with abnormal kidney structure, and 1 GH with glomerulonephritis in childhood). However, all of these women had normal serum levels of uric acid during the follow-up study. If abnormal serum uric acid levels were found during our follow-up study, the cardiologist, who has subsequently been taking care of the patient, ensured further kidney examination. However, we have no data available resulting from this examination yet. We found out that increased uric acid levels were identified mainly in overweight/obese, hypertensive women or in women with simultaneously increased serum levels of homocysteine, CRP, or fasting lipoproteins.

Our results also support the data of other investigators [24,44], who also observed higher lipoprotein A levels among patients with a history of PE or severe PE (PE w/SF), although there is no definite consensus on the role of lipoprotein A in patients with a history of gestational hypertensive disorders involving preeclampsia and intrauterine growth restriction [28,45,46]. Lipoprotein A is a risk factor for coronary heart disease, cardiovascular diseases, atherosclerosis, thrombosis, and stroke [47]. It increases a cardiovascular risk through the mediation of protrombotic and antifibrinolytic effects via competition with plasminogen for its binding site, and stimulation of secretion of plasminogen activator inhibitor-1 [48,49]. Furthermore, lipoprotein A transports the more atherogenic proinflammatory, oxidized phospholipids, which leads to the attraction of inflammatory cells to vessel walls, and proliferation of smooth muscle cell proliferation [50,51,52].

Furthermore, our study indicated no statistical significance in heart rate [33,53,54], and levels of total serum cholesterol levels [21,29,31,39,45,55], serum HDL cholesterol levels [21,29,31,45,55], serum LDL cholesterol levels [21,29,31,55], serum triglycerides levels [21,29,45,55], serum CRP levels [30,45], and plasma homocysteine levels between normal pregnancies and those complicated by either GH or PE, which is in compliance with the data resulting from some previous studies. However, controversial results on serum biomarkers were reported in patients with a history of pregnancy-related complications. Some studies demonstrated altered levels of serum lipids [22,24,28,53], CRP [22,55] or homocysteine [32,56] in women with a prior exposure to gestational hypertensive disorders.

In addition, no studied parameters and serum/plasma biomarkers were altered in women with a history of FGR. These data are in agreement with the data of another study [34], which observed no increased predicted 10-year risk of cardiovascular diseases in women with a history of small for gestational age fetuses (SGA). Nevertheless, our data contradict the findings of other researchers, who observed that a history of low birth weight offspring was associated with an increased cardiovascular risk in women 50 years of age [35], and higher levels of total cholesterol [46]. 

This study may again support the idea that women with hypertensive pregnancy disorders need to be stratified as soon as possible to prevent them from later development of cardiovascular diseases. 

## 4. Materials and Methods 

### 4.1. Participants 

The study was prospective, designed to run from 2016–2018. All women identified as being of white European ethnicity who delivered in the Institute for the Care of the Mother and Child, Prague, Czech Republic within 2007–2013 and were registered in the computerized hospital database (FONSAKORD, STAPRO, Czech Republic) with gestational hypertension (GH), preeclampsia (PE) and/or fetal growth restriction (FGR) were invited to participate in the study. Altogether, 974 women (164 GH cases, 307 PE cases, 147 FGR cases, and 356 NTP cases) were invited to participate in the study. Finally, 276 women (50 GH cases, 102 PE cases, 34 FGR cases, and 90 NTP cases) participated in the study, which represents 28.34% recruitment rate. In the Czech Republic, most of the population identifies as being of white European ethnicity. Ethnic minorities living in Czech Republic represented 0.4% of the population (demographic data collected in 2011 during census). Only a few women from these ethnic minority groups living in Czech Republic were examined and delivered in the Institute for the Care of the Mother and Child, Prague within the period when the study was held. Therefore, they were not included in the study.

Finally, the case cohort included white women with GH (*n* = 50), clinically established PE w or w/o FGR (*n* = 102), FGR (*n* = 34), and 90 women with normal course of gestation that were chosen on the basis of equal maternal age. In-person visit was conducted 3–11 years after the pregnancy ended. Of the 102 patients with preeclampsia, 25 had symptoms of preeclampsia w/o SF and 77 were diagnosed with preeclampsia w/SF. 36 preeclamptic patients required delivery before 34 weeks of gestation and 66 patients delivered after 34 weeks of gestation. Preeclampsia occurred both in previously normotensive patients (98 cases) and was superimposed on pre-existing hypertension (4 cases). Ten growth-retarded fetuses were delivered before 32 weeks of gestation and 24 after 32 weeks of gestation. Oligohydramnios or anhydramnios were present in 19 growth-restricted fetuses and 18 PE cases.

Arterial Doppler examination showed an abnormal pulsatility index (PI) in the umbilical artery (PI > 95th percentile) in 8 PE and 19 FGR cases and in the middle cerebral artery (PI < 5th percentile) in 7 PE and 7 FGR cases. The cerebro-placental ratio (CPR) was <5th percentile in 16 PE and 21 FGR cases. The umbilical artery Doppler showed absent and/or zero diastolic flow in 2 PE and 3 FGR cases. The mean PI in the uterine artery >95th percentile was identified in 13 PE and 8 FGR pregnancies with the presence of unilateral or bilateral diastolic notch in 13 PE and 7 FGR cases. Ductus venosus examination revealed an absence of flow during atrial contraction (a wave) (deep a wave) in 1 FGR pregnancy. In addition, abnormal PI of ductus venosus (>1) was detected in 3 PE and/or FGR pregnancies.

The clinical characteristics of normal and complicated pregnancies are presented in Table 1.

Women with normal pregnancies were defined as those without medical, obstetrical, or surgical complications at the time of the study and who subsequently delivered full term, singleton healthy infants weighing > 2500 g after 37 completed weeks of gestation. Gestational hypertension was defined as high blood pressure that developed after the twentieth week of pregnancy.

Preeclampsia was defined as blood pressure > 140/90 mmHg in two determinations 4 h apart that was associated with proteinuria > 300 mg/24 h after 20 weeks of gestation in a woman with a previously normal blood pressure [57]. In the absence of proteinuria, any of the following can establish the diagnosis of PE: (1) new-onset thrombocytopenia, (2) impaired liver function, (3) renal insufficiency, (4) pulmonary edema, or (5) visual or cerebral disturbances [14]. Severe features of preeclampsia were diagnosed by the presence of one or more of the following findings: (1) systolic blood pressure >160 mmHg or a diastolic blood pressure >110 mmHg, (2) thrombocytopenia (platelet count less than 100,000/microliter), (3) impairment of liver function as indicated by abnormally elevated blood concentrations of liver enzymes (to twice normal concentration), severe persistent right upper quadrant or epigastric pain unresponsive to medication and not accounted for by alternative diagnoses, or both, (4) progressive renal insufficiency (serum creatinine concentration greater than 1.1 mg/dL or a doubling of the serum creatinine concentration in the absence of other renal disease), (5) pulmonary oedema, (6) new onset cerebral or visual disturbances [14]. 

Fetal growth restriction was diagnosed when the estimated fetal weight (EFW), calculated using the Hadlock formula (Astraia, version 1.25.2, Astraia Software GmbH, München, Deutschland), was below the tenth percentile for the evaluated gestational age, adjustments were made for the appropriate population standards of the Czech Republic. In addition to fetal weight below the threshold of the 10th percentile IUGR fetuses had at least one of the following pathological finding: an abnormal pulsatility index in the umbilical artery, absent or reversed end-diastolic velocity waveforms in the umbilical artery, an abnormal pulsatility index in the middle cerebral artery, a sign of a blood flow centralisation, and a deficiency of amniotic fluid (anhydramnios and oligohydramnios). 

Centralization of the fetal circulation represents a protective reaction of the fetus against hypoxia that manifests itself in redistribution of the circulation in the brain, liver and heart at the expense of the flow reduction in the periphery [58,59]. The cerebroplacental ratio (CPR) quantifies redistribution of cardiac output by dividing Doppler indices from representative cerebral and fetoplacental vessels. 

Patients with a complicated gestation demonstrating premature rupture of membranes, in utero infections, fetal anomalies or chromosomal abnormalities, and fetal demise in utero or stillbirth were excluded from the study. 

Written informed consent was provided for all participants included in the study. The study was approved by the Ethics Committee of the Institute for the Care of the Mother and Child, Prague, Czech Republic (grant no. AZV 16-27761A, Long-term monitoring of complex cardiovascular profile in the mother, fetus and offspring descending from pregnancy-related complications, date of approval: 28.5.2015) and by the Ethics Committee of the Third Faculty of Medicine, Prague, Czech Republic (grant no. AZV 16-27761A, Long-term monitoring of complex cardiovascular profile in the mother, fetus and offspring descending from pregnancy-related complications, date of approval: 27.3.2014).

### 4.2. Blood Pressure Measurements

Standardized blood pressure measurements were performed following New AHA Recommendations for Blood Pressure Measurement [60]. Blood pressure was measured three times in the right arm after a five-minute rest period during which the participant sits using an automated device (OMRON M6W, Omron Healthcare Co., Kyoto, Japan). The average of the last two systolic and diastolic pressures was used for the data analyses. 

### 4.3. BMI and Waist Circumference Measurements

Body weight was measured to the nearest 0.05–0.1 kg using an electronic scale and height was measured to the nearest 0.1 cm using a built-in stadiometer (calibrated balance scales, RADWAG WPT 100/200 OW, RADWAG, Czech Republic). BMI was calculated as weight divided by height squared. Waist circumference was measured to the nearest 0.1 cm using a measuring tape. Each measurement was taken twice. Regardless of the height or build, for most adults a waist measurement of greater than 80 cm for women is an indicator of the level of internal fat deposits which coat the heart, kidneys, liver, digestive organs and pancreas. This can increase the risk of heart disease and stroke.

### 4.4. Biological Sampling

Fasting blood samples (from 11 h or more) were collected at the time of the study visit. Total serum cholesterol, high-density lipoprotein (HDL) cholesterol, low-density lipoprotein (LDL) cholesterol, triglycerides, and lipoprotein A Lp(a) were analysed using standard laboratory methods at the Institute for the Care of the Mother and Child. Epidemiological evidence indicates that elevated Lp(a) levels are associated with and increased cardiovascular disease/coronary heart disease risk [61].

Additional outcome measures included high-sensitivity C-reactive protein (CRP), a measure of inflammation, homocysteine, and uric acid. A recent study found that elevated levels of CRP (above 3.0 mg/L led to a greater risk of a coronary disease, heart attack, stroke and probability of having a cardiac procedure, like angioplasty or bypass surgery [62]. 

Elevated levels of homocysteine in the blood have been linked with a wide range of health disorders including heart disease, stroke, macular degeneration, migraine, dementia, cancer, and osteoporosis. Hyperuricemia may be associated with an increase in risk factors for cardiovascular diseases [41]. 

### 4.5. Estimation of Individual and Relative Risks of Having a Heart Attack or Stroke Over the Next Ten Years

We used the QRISK^®^2-2016 Web Calculator (https://qrisk.org/2016/index.php) to work out participant’s risk of having a heart attack or stroke over the next ten years. The QRISK^®^2 score contains many of the traditional risk factors included in Framingham score, but also contains important additional risk factors. The QRISK^®^2 algorithm has been developed by doctors and academics working in the UK National Health Service and is based on routinely collected data such as age, gender, ethnicity, smoking and diabetes status, angina or heart attack in a 1st degree relative below the age of 60, chronic kidney disease (stage 4 or 5), atrial fibrillation, on blood pressure treatment, rheumatoid arthritis, cholesterol/HDL ratio, systolic blood pressure (mmHg), and body mass index calculated on the basis of height (cm) and weight (kg). QRISK^®^2 is a well-established cardiovascular disease risk score, which was designed to identify people at high risk of developing cardiovascular disease who need to be recalled and assessed in more detail to reduce their risk of developing cardiovascular disease. Individual risk of having a heart attack or stroke within the next 10 years is estimated. Parallel, the score of a healthy person with no adverse clinical indicators, normal systolic blood pressure, BMI and a cholesterol ratio corresponding with the age, sex and ethnic group of particular study participant is calculated. Subsequently, a relative cardiovascular disease risk of the study participant is determined. Relative risk is the individual risk divided by the healthy person’s risk. QRISK^®^2-2016 Web Calculator can be used for individuals aged between 25 and 84 years unless they have had a heart attack, angina, stroke, or transient ischemic attack. 

We have not selected the Framingham equations for our follow-up study, since it was documented that they overestimate risk in contemporary European populations by around 100% in Southern European populations and by 50% or more in Northern European populations. Conversely, such equations may underestimate risk in populations such as people with diabetes, South Asian men or the most socially deprived who are at higher than average risk. Overall the Framingham risk equation is likely to overestimate risk in the current Northern European population. They may also underestimate risk in people with extreme risk factor levels or other clinical risks not included in the model (https://qrisk.org/2016/index.php). We believe that the same limitations will be also valid for the Central European population, including the Czech Republic.

### 4.6. Statistical Analysis

Analysis of covariance (ANCOVA), a general linear model which blends ANOVA and regression, was used to test possible differences in mean concentrations of dependant variables (BMI and waist circumference values, the average values of systolic and diastolic blood pressures and heart rate, total serum cholesterol levels, serum HDL cholesterol levels, serum LDL cholesterol levels, serum triglycerides levels, serum Lp(a) levels, serum CRP levels, plasma homocysteine levels, serum uric acid levels, individual and relative risks of having a heart attack or stroke over the next ten years) between groups of patients with normal and complicated course of gestation with respect to particular pregnancy-related disorder subtypes, and the disease severity. 

We made adjustment for potential covariates, where appropriate, including current maternal age, BMI and parity, the time after the birth (in months), the oral contraceptive use status (ex-user, current user, non-user), the smoking status (ex-smoker, smoker, non-smoker), the average values of systolic and diastolic blood pressures, and hypertension on the treatment (yes, no). Prior to the ANCOVA test, Levene’s test for equality of variances was performed (MedCalc, version 16.8.4, MedCalc Software bvba, Ostend, Belgium). The ANCOVA test was applied only if the Levene’s test was negative (*p* > 0.05), which means that the groups were homogeneous. Homogeneity of regression slopes was also tested prior to the performance of ANCOVA test (MedCalc, version 16.8.4, MedCalc Software bvba, Ostend, Belgium). The interpretation of ANCOVA and the associated adjusted means relies on the assumption of homogeneous regression slopes for the various groups [63]. If this assumption is met (*p* > 0.05) then the ANCOVA results are reliable. If the assumptions for ANCOVA were not met (the variances in the groups were different or the assumption of homogeneous regression slopes was not met), the box cox transformation of a dependent variable was applied. The box cox transformation allows create a new variable containing a power transformation of a numeric variable. The transformation is defined by a power parameter λ (Lambda) (MedCalc, version 16.8.4, MedCalc Software bvba, Ostend, Belgium). Alternatively, *z*-scores (a measure indicating how many standard deviations below or above the population mean a raw score is in a normally distributed population) were used for normalization of the data (MedCalc, version 16.8.4, MedCalc Software bvba, Ostend, Belgium).

The statistical analysis performed provided the estimated marginal means with standard error and 95% Confidence Interval of dependent variables for independent variables (pregnancy-related complication groups) (MedCalc, version 16.8.4, MedCalc Software bvba, Ostend, Belgium). In tables outlined in the manuscript, the estimated marginal means with standard errors are reported. Subsequently, pairwise comparisons of estimated marginal means between particular groups was performed. The differences between the estimated marginal means, standard errors, and Bonferroni corrected P-values, and 95% Confidence Intervals of the differences were calculated (MedCalc, version 16.8.4, MedCalc Software bvba, Ostend, Belgium). In tables outlined in the manuscript, Bonferroni corrected P-values are reported.

The unadjusted and adjusted receivers operating characteristic (ROC) curves were constructed to calculate the area under the curve (AUC) and the best cut-off points for particular studied parameters or biomarkers were used in order to calculate the respective sensitivity at 90.0% specificity (MedCalc, version 16.8.4, MedCalc Software bvba, Ostend, Belgium). In ROC analysis, sensitivity expresses the probability that a test result will be positive when the disease was present (true positive rate). Specificity then expresses the probability that a test result will be negative when the disease was not present (true negative rate). We also reported in the tables for unadjusted data the information showing the sensitivity at criterions exceeding the critical values (number of cases exceeding the critical values). 

If the area under the unadjusted ROC curve was significantly different (*p* < 0.05) then we also performed the covariate-specific ROC curve analysis (the studied parameter or biomarker with covariates’ adjustment) using the predicted probabilities gathered from logistic regression. The logistic regression procedure allows to analyse the relationship between one dichotomous dependent variable and one or more independent variables. Another method to evaluate the logistic regression model makes use of ROC curve analysis. In this analysis, the power of the model’s predicted values to discriminate between positive and negative cases is quantified by the Area under the ROC curve (AUC). To perform a full ROC curve analysis the predicted probabilities are first saved and next used as a new variable in ROC curve analysis. The dependent variable used in logistic regression then acts as the classification variable in the ROC curve analysis dialog box.

The unadjusted ROC curve pools all the heterogeneous subjects together and makes an implicit assumption that the decision threshold for test positivity is constant for everybody. The covariate-specific ROC curve is frequently reported in the diagnostic studies, which examines the accuracy of a parameter/biomarker within a subpopulation stratified by the covariates [64]. 

Results were expressed as mean±SE (standard error). The significance level was established at *p*-value of *p* < 0.05.

## 5. Conclusions

We arranged for all patients undergoing this follow-up study the examination in Dpt. of Cardiology, where ECG, echocardiography, functional vascular health assessment for both large and small arteries (EndoPAT), etc. has been performing. After the collection of the data from all patients involved in the follow-up study, we will get a comprehensive overview, how high is a global cardiovascular risk in women after pregnancy-related complications. Nevertheless, from this part of the follow-up study, we can see that a cardiovascular risk is even higher in women after pregnancy-related complications than in women after normal course of gestation, and we can support the idea that the women after pregnancy-related complications might benefit from dispensarisation at primary care physicians or cardiologists, careful monitoring and controlling of risk factors through the implementation of early primary prevention strategies [19]. Healthcare professionals who meet women for the first time later in their lives should take a careful and detailed history of pregnancy complications with focused questions about a history of gestational diabetes mellitus, preeclampsia, preterm birth, or birth of an infant small for gestational age [19].

According to Guideline from the American Heart Association [19] at least following parameters (BMI, waist circumference, blood pressure, and Framingham risk assessment) and serum biomarkers (fasting lipoproteins and glucose) should be checked regularly. 

Our study indicated that the list of examination recommended by AHA may be expanded for the further parameters (heart rate, and extended relative cardiovascular risk score of having a heart attack or stroke over the next ten years using for example QRISK^®^2-2016 Web Calculator, or its updated version QRISK^®^3-2018 Web Calculator) and serum biomarkers (uric acid, and lipoprotein A) in women after pregnancy-related complications and an effort to give them back to normal values should be made.

## Figures and Tables

**Table 1 jcm-08-00544-t001:** Characteristics of cases and controls.

	Normotensive Term Pregnancies (*n* = 90)	PE (*n* = 102)	FGR (*n* = 34)	GH (*n* = 50)	*p*-Value ^1^	*p*-Value ^2^	*p*-Value ^3^
**Pre-existing cardiovascular risk factors before gestation**							
DM type I	0 (0%)	1 (0.98%)	0 (0%)	1 (2.0%)	-	-	-
DM type II	0 (0%)	0 (0%)	1 (2.94%)	0 (0%)	-	-	-
Rheumatoid arthritis	0 (0%)	0 (0%)	1 (2.94%)	2 (4.0%)	-	-	-
Angina or heart attack in a first degree relative before the age of 60 years	2 (2.22%)	0 (0%)	0 (0%)	1 (2.0%)	-	-	-
On blood pressure treatment	0 (0%)	7 (6.86%)	1 (2.94%)	0 (0%)	-	-	-
Hypercholesterolemia	0 (0%)	0 (0%)	0 (0%)	1 (2.0%)	-	-	-
Dispensarisation at Dpt. of Cardiology (valve problems and heart defects)	0 (0%)	1 (0.98%) Sinus tachycardia	1 (2.94%) Leaky heart valve	1 (2.0%) Mitral valve prolapse	-	-	-
Chronic venous insufficiency	0 (0%)	0 (0%)	0 (0%)	1 (2%)	-	-	-
Thrombosis	0 (0%)	2 (1.96%)	0 (0%)	0 (0%)	-	-	-
Presence of risk factors for chronic kidney disease	0 (%)	1 (0.98%) Haematuria	0 (0%)	3 (6.0%) Abnormal kidney structure (*n* = 2) Glomerulonephritis in childhood (*n* = 1)	-	-	-
Chronic kidney disease	0 (%)	1 (0.98%) Nephrotic syndrome	0 (0%)	0 (0%)	-	-	-
**At follow-up**							
Age (years)	38.33 ± 0.45	38.05 ± 0.42	37.0 ± 0.74	39.2 ± 0.61	1.000	0.765	1.000
Time elapsed since delivery (years)	5.73 ± 0.21	5.33 ± 0.20	5.05 ± 0.35	5.06 ± 0.28	1.000	0.610	0.374
Glucose status					0.654	0.346	0.096
Normal	86 (95.56%)	96 (94.12%)	31 (91.18%)	44 (88.00%)			
DM/GDM	4 (4.44%)	6 (5.88%)	3 (8.82%)	6 (12.00%)			
Smoking					0.999	0.619	0.941
Non-Smoker	55 (61.11%)	62 (61.39%)	24 (70.59%)	32 (64.00%)			
Ex-smoker	21 (23.33%)	24 (23.53%)	6 (17.65%)	11 (22.00%)			
Smoker	14 (15.56%)	16 (15.69%)	4 (11.76%)	7 (14.00%)			
Hormonal contraceptive use					0.086	0.379	0.248
No	37 (41.11%)	30 (29.41%)	10 (29.41%)	18 (36.00%)			
In the past	31 (34.44%)	51 (50.00%)	16 (47.06%)	24 (48.00%)			
Yes	22 (24.44%)	21 (20.59)	8 (23.53%)	8 (16.00%)			
Total number of pregnancies per patient					**<0.001**	**0.007**	0.169
1	8 (8.89%)	29 (28.43%)	9 (26.47%)	10 (20.00%)			
2	16 (51.11%)	42 (41.18%)	16 (47.06%)	19 (38.00%)			
3+	36 (40.00%)	31 (30.39%)	9 (26.47%)	21 (42.00%)			
Total parity per patient					**<0.001**	**0.046**	**0.025**
1	13 (14.44%)	40 (39.22%)	11 (32.36%)	17 (34.00%)			
2	63 (70.00%)	52 (50.98%)	21 (61.76%)	27 (54.00%)			
3+	14 (15.56%)	10 (9.80%)	2 (5.71%)	6 (12%)			
**During gestation**							
Maternal age at delivery (years)	32.64 ± 0.42	32.54 ± 0.40	31.91 ± 0.69	34.08 ± 0.57	1.000	1.000	0.274
GA at delivery (weeks)	39.91 ± 0.28	35.97 ± 0.27	35.65 ± 0.46	38.73 ± 0.38	**<0.001**	**<0.001**	0.987
Fetal birth weight (g)	3402.44 ± 70.01	2416.74 ± 65.76	1910.00 ± 113.91	3242.40 ± 93.93	**<0.001**	**<0.001**	1.000
Mode of delivery					**<0.001**	**<0.001**	**<0.001**
Vaginal	83 (92.22%)	19 (18.63%)	7 (20.59%)	22 (44.00%)			
CS	7 (7.78%)	83 (81.37%)	27 (79.41%)	28 (56.00%)			
Fetal sex					0.315	0.740	0.405
Boy	48 (53.33%)	47 (46.08%)	17 (50.00%)	23 (54.00%)			
Girl	42 (46.67%)	55 (53.92%)	17 (50.00%)	27 (46.00%)			
Blood pressure (mmHg)							
Systolic	120.57 ± 1.51	158.38 ± 1.42	128.08 ± 2.44	147.79 ± 2.03	**<0.001**	0.092	**<0.001**
Diastolic	75.77 ± 1.03	98.74 ± 0.97	79.26 ± 1.66	93.89 ± 1.38	**<0.001**	0.451	**<0.001**
Infertility treatment					**0.027**	**0.004**	0.096
Yes	4 (4.44%)	14 (13.73%)	7 (20.59%)	6 (12.00%)			
No	86 (95.65%)	88 (86.27%)	27 (79.41%)	44 (88.00%)			

Data are presented as mean ± SE (standard error) for continuous variables and as number (percent) for categorical variables. Statistically significant results are marked in bold. Continuous variables were compared using ANOVA test. Categorical variables were compared using Chi-squared test. *p*-value ^1, 2, 3^: the comparison among normal pregnancies and preeclampsia, fetal growth restriction or gestational hypertension, respectively. Categorical variables were compared using a chi-square test; PE, preeclampsia; GH, gestational hypertension; FGR, fetal growth restriction; GA, gestational age; CS, Caesarean section.

**Table 2 jcm-08-00544-t002:** Impact of GH or PE history on maternal cardiovascular risk-overview (ANOVA and ANCOVA analyses).

		NTP (*n* = 90)	FGR (*n* = 34)	GH (*n* = 50)	PE (*n* = 102)	Diagnostic Groups (Normal vs Diseased)	*p* Value (ANOVA, ANCOVA)
Serum uric acid (μmol/L)	Unadjusted data	248.044 (6.212)	278.529 (10.051)	286.081 (8.372)	276.792 (5.832)	NTP vs ↑FGR	*p* = 0.094
						**NTP vs ↑ GH**	***p* = 0.004**
						**NTP vs ↑ PE**	***p* = 0.005**
BMI	Unadjusted data	23.100 (0.517)	23.928 (0.941)	27.228 (0.694)	25.946 (0.486)	**NTP vs ↑ GH**	***p* < 0.001**
						**NTP vs ↑ PE**	***p* < 0.001**
	Adjusted data	23.138 (0.525) ^A^	24.529 (0.860) ^A^	26.987 (0.694) ^A^	25.945 (0.490) ^A^	**NTP vs ↑ GH**	***p* = 0.002**
						**NTP vs ↑ PE**	***p* = 0.001**
Waist circumference (cm)	Unadjusted data	76.605 (1.277)	78.264 (2.078)	86.770 (1.713)	83.852 (1.199)	**NTP vs ↑ GH**	***p* < 0.001**
						**NTP vs ↑ PE**	***p* < 0.001**
	Adjusted data	76.891 (1.298) ^A^	79.546 (2.128) ^A^	86.118 (1.716) ^A^	83.624 (1.212) ^A^	**NTP vs ↑ GH**	***p* = 0.001**
						**NTP vs ↑ PE**	***p* = 0.001**
SBP (mmHg)	Unadjusted data	112.911 (1.316)	118.088 (2.142)	129.580 (1.766)	123.656 (1.236)	**NTP vs ↑ GH**	***p* < 0.001**
						**NTP vs ↑ PE**	***p* < 0.001**
	Adjusted data	114.895 (1.286) ^A^	118.325 (2.058) ^A^	127.517 (1.683) ^A^	122.642 (1.178) ^A^	**NTP vs ↑ GH**	***p* < 0.001**
						**NTP vs ↑ PE**	***p* = 0.029**
DBP (mmHg)	Unadjusted data	72.100 (1.000)	77.058 (1.627)	82.940 (1.342)	79.490 (0.939)	**NTP vs ↑ GH**	***p* < 0.001**
						**NTP vs ↑ PE**	***p* < 0.001**
	Adjusted data	73.898 (0.977) ^A^	77.091 (1.564) ^A^	81.295 (1.279) ^A^	78.737 (0.895) ^A^	**NTP vs ↑ GH**	***p* < 0.001**
						**NTP vs ↑ PE**	***p* = 0.002**
Heart rate (bpm)	Unadjusted data	71.644 (1.079)	73.969 (1.782)	76.300 (1.447)	72.627 (1.013)	NTP vs ↑GH	*p* = 0.062
	Adjusted data	71.705 (1.141) ^A^	74.294 (1.861) ^A^	76.262 (1.497) ^A^	72.670 (1.048) ^A^	NTP vs ↑GH	*p* = 0.119
Relative QRISK^®^2 risk score	Unadjusted data	0.920 (0.134)	1.379 (0.216)	1.966 (0.178)	1.617 (0.125)	**NTP vs ↑ GH**	***p* < 0.001**
						**NTP vs ↑ PE**	***p* < 0.001**
	Adjusted data	0.865 (0.118) ^A^	1.420 (0.195) ^A^	1.984 (0.155) ^A^	1.578 (0.111) ^A^	**NTP vs ↑ GH**	***p* < 0.001**
						**NTP vs ↑ PE**	***p* < 0.001**

Data are expressed as mean (SE; standard error). Analysis of variance (ANOVA) was used for unadjusted data and Analysis of covariance (ANCOVA) for adjusted data. The significance level was established at *p*-value of *p* < 0.05 (Bonferroni corrected *p*-values). ^A^ Adjusted for potential covariates, where appropriate, including current maternal age, BMI, parity, the time after the birth (in months), the oral contraceptive use status (ex-user, current user, non-user), the smoking status (ex-smoker, smoker, non-smoker), the average values of systolic and diastolic blood pressures, and hypertension on the treatment (yes, no). FGR, fetal growth restriction; GH, gestational hypertension; PE, preeclampsia; NTP, normotensive term pregnancies; BMI, body mass index; SBP, systolic blood pressure; DBP, diastolic blood pressure; bpm, beats per minute.

**Table 3 jcm-08-00544-t003:** Impact of GH or PE history on maternal cardiovascular risk - overview (Receive operating characteristics (ROC) curve analysis).

		Diagnostic Groups (Normal vs Diseased)	ROC Curve Parameters	Sensitivity at 10% FPR	Sensitivity and Specificity When Critical Values are Exceeded
Serum uric acid (μmol/L)	Unadjusted data	NTP vs FGR	AUC 0.615, *p* = 0.056	32.35% Criterion > 308.5 μmol/L	26.46% sensitivity at 97.5% specificity Criterion > 340.55 μmol/L (hyperuricemia)
		NTP vs GH	AUC 0.670, *p* < 0.001	36.53% Criterion > 308.5 μmol/L	14.29% sensitivity at 97.5% specificity Criterion > 340.55 μmol/L (hyperuricemia)
		NTP vs PE	AUC 0.644, *p* < 0.001	28.51% Criterion > 311.2 μmol/L	13.09% sensitivity at 97.5% specificity Criterion > 340.775 μmol/L (hyperuricemia)
BMI	Unadjusted data	NTP vs GH	AUC 0.738, *p* < 0.001	42.00% Criterion > 27.78 (overweight)	20.0% sensitivity at 97.5% specificity Criterion > 31.02 (obese class I, moderately obese)
		NTP vs PE	AUC 0.670, *p* < 0.001	27.45% Criterion > 27.78 (overweight)	18.63% sensitivity at 97.5% specificity Criterion > 31.02 (obese class I, moderately obese)
	Adjusted data	NTP vs GH	AUC 0.899, *p* < 0.001	74.00%	-
		NTP vs PE	AUC 0.791, *p* < 0.001	53.06%	-
Waist circumference (cm)	Unadjusted data	NTP vs GH	AUC 0.743, *p* < 0.001	42.00% Criterion > 87 cm	36.0% sensitivity at 91.11% specificity Criterion > 88 cm (obese, high cardiovascular risk)
		NTP vs PE	AUC 0.688, *p* < 0.001	32.35% Criterion > 87 cm	29.41% sensitivity at 91.11% specificity Criterion > 88 cm (obese, high cardiovascular risk)
	Adjusted data	NTP vs GH	AUC 0.902, *p* < 0.001	70.00%	-
		NTP vs PE	AUC 0.796, *p* < 0.001	54.08%	-
SBP (mmHg)	Unadjusted data	NTP vs GH	AUC 0.843, *p* < 0.001	54.00% Criterion > 123.4 mmHg (prehypertension)	18.00% sensitivity at 100.0% specificity Criterion > 141 mmHg (hypertension)
		NTP vs PE	AUC 0.750, *p* < 0.001	46.86% Criterion > 123.4 mmHg (prehypertension)	10.78% sensitivity at 100.0% specificity Criterion > 141 mm Hg (hypertension)
	Adjusted data	NTP vs GH	AUC 0.822, *p* < 0.001	62.00%	-
		NTP vs PE	AUC 0.754, *p* < 0.001	51.10%	-
DBP (mmHg)	Unadjusted data	NTP vs GH	AUC 0.794, *p* < 0.001	51.00% Criterion > 80.5 mmHg (prehypertension)	20.0% sensitivity at 100.0% specificity Criterion > 91 mmHg (hypertension)
		NTP vs PE	AUC 0.714, *p* < 0.001	39.71% Criterion > 80.5 mmHg (prehypertension)	12.75% sensitivity at 100.0% specificity Criterion > 91 mmHg (hypertension)
	Adjusted data	NTP vs GH	AUC 0.875, *p* < 0.001	60.00%	-
		NTP vs PE	AUC 0.778, *p* < 0.001	47.96%	-
Heart rate (bpm)	Unadjusted data	NTP vs GH	AUC 0.619, *p* = 0.017	18.00% Criterion >84 bpm	4.0% sensitivity at 100.0% specificity Criterion > 107 bpm (tachycardia)
	Adjusted data	NTP vs GH	AUC 0.833, *p* < 0.001	54.00%	-
Relative QRISK^®^2 risk score	Unadjusted data	NTP vs GH	AUC 0.789, *p* < 0.001	30.00% Criterion > 1.60	18.0% sensitivity at 100.0% specificity Criterion > 2.9
		NTP vs PE	AUC 0.711, *p* < 0.001	26.37% Criterion > 1.60	12.75% sensitivity at 100.0% specificity Criterion > 2.9
	Adjusted data	NTP vs GH	AUC 0.894, *p* < 0.001	74.00%	-
		NTP vs PE	AUC 0.788, *p* < 0.001	55.10%	-

The unadjusted and adjusted receivers operating characteristic (ROC) curves were constructed to calculate the area under the curve (AUC) and the best cut-off points for particular studied parameters or biomarkers were used in order to calculate the respective sensitivity at 90.0% specificity (MedCalc, version 16.8.4, MedCalc Software bvba, Ostend, Belgium). We also reported for unadjusted data the information showing the sensitivity at criterions exceeding the critical values (number of cases exceeding the critical values). Data were adjusted for potential covariates, where appropriate, including current maternal age, BMI, parity, the time after the birth (in months), the oral contraceptive use status (ex-user, current user, non-user), the smoking status (ex-smoker, smoker, non-smoker), the average values of systolic and diastolic blood pressures, and hypertension on the treatment (yes, no). The significance level was established at *p*-value of *p* < 0.05. FGR, fetal growth restriction; GH, gestational hypertension; PE, preeclampsia; NTP, normotensive term pregnancies; BMI, body mass index; SBP, systolic blood pressure; DBP, diastolic blood pressure; bpm, beats per minute; AUC, area under curve; FPR, false positive rate.

**Table 4 jcm-08-00544-t004:** Impact of severity of PE on maternal cardiovascular risk - overview (ANOVA and ANCOVA analyses).

		NTP (*n* = 90)	PE w/o SF (*n* = 25)	PE w/SF (*n* = 77)	Diagnostic Groups (Normal vs Diseased)	*p* Value (ANOVA, ANCOVA)
Serum uric acid (μmol/L)	Unadjusted data	248.044 (5.649)	275.880 (10.659)	277.092 (6.113)	**NTP vs ↑ PE w/SF**	***p* = 0.001**
BMI	Unadjusted data	23.100 (0.480)	26.037 (0.912)	25.916 (0.519)	**NTP vs ↑ PE w/o SF**	***p* = 0.021**
					**NTP vs ↑ PE w/SF**	***p* < 0.001**
	Adjusted data	23.518 (0.479) ^A^	26.300 (0.883) ^A^	25.378 (0.518) ^A^	**NTP vs ↑ PE w/o SF**	***p* = 0.021**
					**NTP vs ↑ PE w/SF**	***p* = 0.015**
Waist circumference (cm)	Unadjusted data	76.605 (1.191)	83.480 (2.260)	83.974 (1.287)	**NTP vs ↑ PE w/o SF**	***p* = 0.010**
					**NTP vs ↑ PE w/SF**	***p* < 0.001**
	Adjusted data	77.796 (1.206) ^A^	83.607 (2.225) ^A^	82.416 (1.304) ^A^	**NTP vs ↑ PE w/o SF**	***p* = 0.029**
					**NTP vs ↑ PE w/SF**	***p* = 0.013**
SBP (mmHg)	Unadjusted data	112.911(1.203)	122.160 (2.283)	124.142 (1.301)	**NTP vs ↑ PE w/o SF**	***p* = 0.001**
					**NTP vs ↑ PE w/SF**	***p* < 0.001**
	Adjusted data	116.949 (0.743) ^A^	120.994 (1.379) ^A^	119.791 (0.810) ^A^	**NTP vs ↑ PE w/o SF**	***p* = 0.055**
					**NTP vs ↑ PE w/SF**	***p* = 0.121**
DBP (mmHg)	Unadjusted data	72.100 (0.934)	77.520 (1.773)	80.129 (1.010)	**NTP vs ↑ PE w/o SF**	***p* = 0.014**
					**NTP vs ↑ PE w/SF**	***p* < 0.001**
	Adjusted data	72.896 (0.948) ^A^	77.841 (1.810) ^A^	79.376 (1.037) ^A^	**NTP vs ↑ PE w/o SF**	***p* = 0.036**
					**NTP vs ↑ PE w/SF**	***p* < 0.001**
Relative QRISK^®^2 risk score	Unadjusted data	0.920 (0.116)	1.228 (0.220)	1.744 (0.125)	**NTP vs ↑ PE w/o SF**	***p* = 0.008**
					**NTP vs ↑ PE w/SF**	***p* < 0.001**
	Adjusted data	1.060 (0.098) ^A^	1.201 (0.182) ^A^	1.553 (0.106) ^A^	**NTP vs ↑ PE w/o SF**	***p* = 0.003**
					**NTP vs ↑ PE w/SF**	***p* < 0.001**

Data are expressed as mean (SE; standard error). Analysis of variance (ANOVA) was used for unadjusted data and Analysis of covariance (ANCOVA) for adjusted data. The significance level was established at *p*-value of *p* < 0.05 (Bonferroni corrected *p*-values). ^A^ Adjusted for potential covariates, where appropriate, including current maternal age, BMI, parity, the time after the birth (in months), the oral contraceptive use status (ex-user, current user, non-user), the smoking status (ex-smoker, smoker, non-smoker), the average values of systolic and diastolic blood pressures, and hypertension on the treatment (yes, no). PE w/o SF, preeclampsia without severe features; PE w/SF, preeclampsia with severe features; NTP, normotensive term pregnancies; BMI, body mass index; SBP, systolic blood pressure; DBP, diastolic blood pressure.

**Table 5 jcm-08-00544-t005:** Impact of severity of PE on maternal cardiovascular risk - overview (ROC curve analysis).

		Diagnostic Groups (Normal vs Diseased)	ROC Curve Parameters	Sensitivity at 10% FPR	Sensitivity and Specificity When Critical Values are Exceeded
Serum uric acid (μmol/L)	Unadjusted data	NTP vs PE w/SF	AUC 0.648, *p* < 0.001	28.82% Criterion > 311.2 μmol/L	13.45% sensitivity at 97.5% specificity Criterion > 340.775 μmol/L (hyperuricemia)
BMI	Unadjusted data	NTP vs PE w/o SF	AUC 0.666, *p* = 0.007	20.00% Criterion > 27.78 (overweight)	20.0% sensitivity at 97.5% specificity Criterion > 31.02 (obese class I, moderately obese)
		NTP vs PE w/SF	AUC 0.672, *p* < 0.001	29.87% Criterion > 27.78 (overweight)	18.18% sensitivity at 97.5% specificity Criterion > 31.02 (obese class I, moderately obese)
	Adjusted data	NTP vs PE w/o SF	AUC 0.853, *p* < 0.001	62.50%	-
		NTP vs PE w/SF	AUC 0.781, *p* < 0.001	56.76%	-
Waist circumference (cm)	Unadjusted data	NTP vs PE w/o SF	AUC 0.702, *p* = 0.001	28.00% Criterion > 87 cm	24.0% sensitivity at 91.11% specificity Criterion > 88 cm (obese, high cardiovascular risk)
		NTP vs PE w/SF	AUC 0.683, *p* < 0.001	33.77% Criterion > 87 cm	31.17% sensitivity at 91.11% specificity Criterion > 88 cm (obese, high cardiovascular risk)
	Adjusted data	NTP vs PE w/o SF	AUC 0.847, *p* < 0.001	58.33%	-
		NTP vs PE w/SF	AUC 0.788, *p* < 0.001	55.41%	-
SBP (mmHg)	Unadjusted data	NTP vs PE w/o SF	AUC 0.740, *p* < 0.001	40.00% Criterion > 123.4 mmHg (prehypertension)	4.00% sensitivity at 100.0% specificity Criterion > 141 mm Hg (hypertension)
		NTP vs PE w/SF	AUC 0.753, *p* < 0.001	49.09% Criterion > 123.4 mmHg (prehypertension)	12.99% sensitivity at 100.0% specificity Criterion > 141 mm Hg (hypertension)
	Adjusted data	NTP vs PE w/o SF	AUC 0.805, *p* < 0.001	62.50%	-
		NTP vs PE w/SF	AUC 0.762, *p* < 0.001	56.76%	-
DBP (mmHg)	Unadjusted data	NTP vs PE w/o SF	AUC 0.669, *p* = 0.002	24.00% Criterion > 80.5 mmHg (prehypertension)	12.00% sensitivity at 100.0% specificity Criterion > 91 mm Hg (hypertension)
		NTP vs PE w/SF	AUC 0.729, *p* < 0.001	44.81% Criterion > 80.5 mmHg (prehypertension)	12.99% sensitivity at 100.0% specificity Criterion > 91 mm Hg (hypertension)
	Adjusted data	NTP vs PE w/o SF	AUC 0.729, *p* < 0.001	45.83%	-
		NTP vs PE w/SF	AUC 0.747, *p* < 0.001	50.00%	-
Relative QRISK^®^2 risk score	Unadjusted data	NTP vs PE w/o SF	AUC 0.723, *p* < 0.001	19.87% Criterion > 1.60	0.0% sensitivity at 100.0% specificity Criterion > 2.9
		NTP vs PE w/SF	AUC 0.707, *p* < 0.001	28.48% Criterion > 1.60	16.88% sensitivity at 100.0% specificity Criterion > 2.9
	Adjusted data	NTP vs PE w/o SF	AUC 0.843, *p* < 0.001	58.33%	-
		NTP vs PE w/SF	AUC 0.782, *p* < 0.001	54.05%	-

The unadjusted and adjusted receivers operating characteristic (ROC) curves were constructed to calculate the area under the curve (AUC) and the best cut-off points for particular studied parameters or biomarkers were used in order to calculate the respective sensitivity at 90.0% specificity (MedCalc, version 16.8.4, MedCalc Software bvba, Ostend, Belgium). We also reported for unadjusted data the information showing the sensitivity at criterions exceeding the critical values (number of cases exceeding the critical values). Data were adjusted for potential covariates, where appropriate, including current maternal age, BMI, parity, the time after the birth (in months), the oral contraceptive use status (ex-user, current user, non-user), the smoking status (ex-smoker, smoker, non-smoker), the average values of systolic and diastolic blood pressures, and hypertension on the treatment (yes, no). The significance level was established at *p*-value of *p* < 0.05. PE w/o SF, preeclampsia without severe features; PE w/SF, preeclampsia with severe features; NTP, normotensive term pregnancies; BMI, body mass index; SBP, systolic blood pressure; DBP, diastolic blood pressure; AUC, area under curve; FPR, false positive rate.

**Table 6 jcm-08-00544-t006:** Impact of PE with respect to delivery date on maternal cardiovascular risk - overview (ANOVA and ANCOVA analyses).

		NTP (*n* = 90)	Early PE (*n* = 36)	Late PE (*n* = 66)	Diagnostic Groups (Normal vs Diseased)	*p* Value (ANOVA, ANCOVA)
Serum Lp(a) (nmol/L)	Unadjusted data	36.449 (7.509)	90.828 (11.974)	45.483 (8.720)	**NTP vs ↑ early PE**	***p* = 0.037**
	Adjusted data	38.909 (8.094) ^A^	88.341 (13.631) ^A^	46.977 (8.921) ^A^	NTP vs early PE	*p* = 0.192
Serum uric acid (μmol/L)	Unadjusted data	248.044 (5.625)	285.971 (8.971)	271.924 (6.532)	**NTP vs ↑ early PE**	***p* = 0.001**
					**NTP vs ↑ late PE**	***p* = 0.018**
BMI	Unadjusted data	23.100 (0.465)	28.085 (0.735)	24.779 (0.543)	**NTP vs ↑ early PE**	***p* < 0.001**
					**NTP vs ↑ late PE**	***p* = 0.034**
	Adjusted data	23.440 (0.475) ^A^	26.948 (0.784) ^A^	24.998 (0.535) ^A^	**NTP vs early PE**	***p* = 0.003**
					**NTP vs ↑ late PE**	***p* = 0.031**
Waist circumference (cm)	Unadjusted data	76.605 (1.161)	88.472 (1.836)	81.333 (1.356)	**NTP vs ↑ early PE**	***p* < 0.001**
					**NTP vs ↑ late PE**	***p =* 0.006**
	Adjusted data	77.654 (1.202) ^A^	85.201 (1.983) ^A^	81.578 (1.353) ^A^	**NTP vs ↑ early PE**	***p* = 0.011**
					**NTP vs ↑ late PE**	***p* = 0.018**
SBP (mmHg)	Unadjusted data	112.911(1.178)	128.055 (1.864)	121.257 (1.376)	**NTP vs ↑ early PE**	***p* < 0.001**
					**NTP vs ↑ late PE**	***p* < 0.001**
	Adjusted data	116.931 (0.746) ^A^	120.196 (1.243) ^A^	120.052 (0.845) ^A^	NTP vs ↑early PE	*p* = 0.099
					**NTP vs ↑ late PE**	***p* = 0.021**
DBP (mmHg)	Unadjusted data	72.100 (0.914)	83.166 (1.446)	77.484 (1.068)	**NTP vs ↑ early PE**	***p* < 0.001**
					**NTP vs ↑ late PE**	***p* < 0.001**
	Adjusted data	72.837 (0.937) ^A^	81.848 (1.554) ^A^	77.570 (1.091) ^A^	**NTP vs ↑ early PE**	***p* < 0.001**
					**NTP vs ↑ late PE**	***p* = 0.004**
Relative QRISK^®^2 risk score	Unadjusted data	0.920 (0.108)	2.436 (0.170)	1.171 (0.125)	**NTP vs ↑ early PE**	***p* < 0.001**
					**NTP vs ↑ late PE**	***p* = 0.009**
	Adjusted data	1.036 (0.093) ^A^	2.099 (0.151) ^A^	1.162 (0.105) ^A^	**NTP vs ↑ early PE**	***p* < 0.001**
					**NTP vs ↑ late PE**	***p* = 0.005**

Data are expressed as mean (SE; standard error). Analysis of variance (ANOVA) was used for unadjusted data and Analysis of covariance (ANCOVA) for adjusted data. The significance level was established at *p*-value of *p* < 0.05 (Bonferroni corrected p-values). ^A^ Adjusted for potential covariates, where appropriate, including current maternal age, BMI, parity, the time after the birth (in months), the oral contraceptive use status (ex-user, current user, non-user), the smoking status (ex-smoker, smoker, non-smoker), the average values of systolic and diastolic blood pressures, and hypertension on the treatment (yes, no). PE, preeclampsia; NTP, normotensive term pregnancies; Lp(a), lipoprotein a; BMI, body mass index; SBP, systolic blood pressure; DBP, diastolic blood pressure.

**Table 7 jcm-08-00544-t007:** Impact of PE with respect to delivery date on maternal cardiovascular risk - overview (ROC curve analysis).

		Diagnostic Groups (Normal vs Diseased)	ROC Curve Parameters	Sensitivity at 10% FPR	Sensitivity and Specificity When Critical Values are Exceeded
Serum Lp(a) (nmol/L)	Unadjusted data	NTP vs early PE	AUC 0.632, *p* = 0.022	31.43% Criterion > 89.26 nmol/L (risk of CVD)	34.29% sensitivity at 86.52% specificity Criterion > 73.20 nmol/L (risk of CVD)
	Adjusted data	NTP vs early PE	AUC 0.905, *p* < 0.001	81.82%	-
Serum uric acid (μmol/L)	Unadjusted data	NTP vs early PE	AUC 0.667, *p* = 0.004	42.86% Criterion > 308.5 μmol/L	20.00% sensitivity at 97.5% specificity Criterion > 340.77 μmol/L (hyperuricemia)
		NTP vs late PE	AUC 0.632, *p* = 0.003	20.91% Criterion > 311.2 μmol/L	9.43% sensitivity at 97.5% specificity Criterion > 340.55 μmol/L (hyperuricemia)
BMI	Unadjusted data	NTP vs early PE	AUC 0.748, *p* < 0.001	38.89% Criterion > 27.78 (overweight)	33.33% sensitivity at 97.5% specificity Criterion > 31.02 (obese class I, moderately obese)
		NTP vs late PE	AUC 0.628, *p* = 0.004	21.21% Criterion >27.78 (overweight)	10.61% sensitivity at 97.5% specificity Criterion > 31.02 (obese class I, moderately obese)
	Adjusted data	NTP vs early PE	AUC 0.894, *p* < 0.001	67.65%	-
		NTP vs late PE	AUC 0.767, *p* < 0.001	48.44%	-
Waist circumference (cm)	Unadjusted data	NTP vs early PE	AUC 0.740, *p* < 0.001	50.00% Criterion > 87 cm	44.44% sensitivity at 91.11% specificity Criterion > 88 cm (obese, high cardiovascular risk)
		NTP vs late PE	AUC 0.659, *p* < 0.001	22.73%Criterion > 87 cm	21.21% sensitivity at 91.11% specificity Criterion > 88 cm (obese, high cardiovascular risk)
	Adjusted data	NTP vs early PE	AUC 0.890, *p* < 0.001	64.71%	-
		NTP vs late PE	AUC 0.775, *p* < 0.001	50.00%	-
SBP (mmHg)	Unadjusted data	NTP vs early PE	AUC 0.859, *p* < 0.001	62.78% Criterion > 123.4 mmHg (prehypertension)	16.67% sensitivity at 100.0% specificity Criterion > 141 mmHg (hypertension)
		NTP vs late PE	AUC 0.690, *p* < 0.001	38.18% Criterion > 123.4 mmHg (prehypertension)	7.58% sensitivity at 100.0% specificity Criterion > 141 mmHg (hypertension)
	Adjusted data	NTP vs early PE	AUC 0.884, *p* < 0.001	70.59%	-
		NTP vs late PE	AUC 0.724, *p* < 0.001	46.88%	-
DBP (mmHg)	Unadjusted data	NTP vs early PE	AUC 0.824, *p* < 0.001	54.17% Criterion > 80.5 mmHg (prehypertension)	16.67% sensitivity at 100.0% specificity Criterion > 91 mmHg (hypertension)
		NTP vs late PE	AUC 0.654, *p* < 0.001	31.82% Criterion > 80.5 mmHg (prehypertension)	10.61% sensitivity at 100.0% specificity Criterion > 91 mmHg (hypertension)
	Adjusted data	NTP vs early PE	AUC 0.874, *p* < 0.001	64.71%	-
		NTP vs late PE	AUC 0.704, *p* < 0.001	34.38%	-
Relative QRISK^®^2 risk score	Unadjusted data	NTP vs early PE	AUC 0.802, *p* < 0.001	41.67% Criterion > 1.60	33.33% sensitivity at 100.0% specificity Criterion > 2.9
		NTP vs late PE	AUC 0.661, *p* < 0.001	18.03% Criterion > 1.60	1.52% sensitivity at 100.0% specificity Criterion > 2.9
	Adjusted data	NTP vs early PE	AUC 0.886, *p* < 0.001	73.53%	-
		NTP vs late PE	AUC 0.749, *p* < 0.001	51.56%	-

The unadjusted and adjusted receivers operating characteristic (ROC) curves were constructed to calculate the area under the curve (AUC) and the best cut-off points for particular studied parameters or biomarkers were used in order to calculate the respective sensitivity at 90.0% specificity (MedCalc, version 16.8.4, MedCalc Software bvba, Ostend, Belgium). We also reported for unadjusted data the information showing the sensitivity at criterions exceeding the critical values (number of cases exceeding the critical values). Data were adjusted for potential covariates, where appropriate, including current maternal age, BMI, parity, the time after the birth (in months), the oral contraceptive use status (ex-user, current user, non-user), the smoking status (ex-smoker, smoker, non-smoker), the average values of systolic and diastolic blood pressures, and hypertension on the treatment (yes, no). The significance level was established at *p*-value of *p* < 0.05. PE, preeclampsia; NTP, normotensive term pregnancies; Lp(a), lipoprotein a; BMI, body mass index; SBP, systolic blood pressure; DBP, diastolic blood pressure; AUC, area under curve; FPR, false positive rate.

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
