# Peer review of "Maternal Cardiovascular Risk Assessment 3-to-11 Years Postpartum in Relation to Previous Occurrence of Pregnancy-Related Complications"

_jcm, 2019, doi:10.3390/jcm8040544_

Reviewer 1 Report

The authors investigate the association between the previous maternal occurrence of pregnancy-related complications and cardiovascular risk of postpartum 3 to 11 years by using QRISK®2 CVD score. QRISK2 is likely to be a more efficient and equitable tool for treatment decisions for the primary prevention of cardiovascular disease (BMJ 2008;336:a332.). If QRISK®2 CVD score is a good medical screening tool. In this study, it is not add a lot of new information.

Comments:

In Tables 1, 2 and 3:

1.

These dependent variables (outcome) include BMI, waist circumference values, SBP, DBP, heart rate, total serum cholesterol, HDL, LDL, triglycerides, lipoprotein A, CRP, homocysteine levels, uric acid, and relative QRISK®2 risk score.

The independent variable is pregnancy-related complication groups of gestational hypertension (GH), preeclampsia (PE) and fetal growth restriction (FGR)).

A general linear model (ANOVA or ANCOVA) was used to test possible differences in mean concentrations of dependent variables between the means of pregnancy-related complication groups.

The authors should be able to do that the post hoc analysis of observed means for unadjusted data and pairwise comparisons of estimated marginal means for adjusted data based on the original scale of dependent variable.

If the authors have an analysis, it should be written in the manuscript (Statistical analysis section).

2.

The dependent variable (outcome) is pregnancy-related complication groups of GH, PE(mild PE, severe PE, early PE or late PE) or FGR (versus NTP, normotensive term pregnancies). 

These independent variables include the main factor of BMI, waist circumference values, SBP, DBP, heart rate, total serum cholesterol, HDL, LDL, triglycerides, lipoprotein A, CRP, homocysteine levels, uric acid, or relative QRISK®2 risk score and the potential covariates.

The authors analyze the relationships between one dichotomous dependent variable (pregnancy-related complication group versus NTP) and by main factor after adjusted for potential covariates using the logistic regression models.

The unadjusted and adjusted receivers operating characteristic (ROC) curves also were used to analyze the relationships using the logistic regression models

 3.

Together(1+2), different dependent variables (outcomes) exist in the same table; it seems easy to confuse readers to read the article.

Author Response

Please see the attachement - Answers to Reviewer 1 Comments

Reviewer 2 Report

Thank you for the opportunity to review this interesting manuscript which explores cardiovascular risk prediction in women with a history of placental disease.  The strengths of the study include its prospective design and comprehensive and detailed data collection.  However I would like to understand in more detail about the clinical implications of the findings.

The authors describe a large number of other studies which have also reported both positive and negative findings but there is no detail of how their cohorts or study design differ which may reflect the current studies findings. 

I am also unclear about the novelty in this application.  It appears that there is detailed prospective data collection for a median of 5 years in study groups but this is not described in detail or highlighted as a study strength.

How does these data assist with future risk prediction – how should postpartum care be adjusted?

Why were ACOG definitions from 2002 used?

Why was the QRISK2 score chosen for a group of young women – with overall low risk? Are there better scores that could be used?  Should a prediction score be developed for this demographic?

Was any analysis conducted to explore whether there is likely to be any confounding due to the self-selection of individuals that attended follow-up study visits.

The tables are complex and difficult to navigate – suggest moving to an appendix and reporting significant findings only.

Suggest presenting Table 1 first – with demographic data. What was the ethnicity of the women?

The sensitivity analysis is unclear – what is being predicted?

The introduction last sentence is difficult to understand – could it be simplified

I would like to see some more clinical interpretation of the data – do these women have pre-existing risk factors or does pregnancy confer additional risk. 

What was renal function – could this explain the uric acid differences between groups?

Were women with pre-existing HT or CKD excluded?

What is the relevance of describing CPR – is this reported elsewhere?

What were recruitment rates?  How many women declined to take part?

Is it possible to compare any of the data with those collected at baseline in pregnancy?

Author Response

Please see the attachment - Answers to Reviewer 2 Comments

Round  2

Reviewer 1 Report

None

Reviewer 2 Report

No further comments.